# Study on Italian Onion Cultivars/Ecotypes towards Onion Yellow Dwarf Virus (OYDV) Infection

Carla Libia Corrado [1,2,†], Giuseppe Micali [3,†], Antonio Mauceri [3], Sabrina Bertin [2], Francesco Sunseri [3], Maria Rosa Abenavoli [3] and Antonio Tiberini [2,*]

1 Department of Environmental Biology, Sapienza University of Rome, Piazzale Aldo Moro, 5, 00185 Rome, Italy; carlalibia.corrado@uniroma1.it or carlalibia.corrado@crea.gov.it
2 Council for Agricultural Research and Economics, Research Centre for Plant Protection and Certification, Via C. G. Bertero, 22, 00156 Rome, Italy; sabrina.bertin@crea.gov.it
3 Department of Agraria, University Mediterranea of Reggio Calabria, Località Feo di Vito snc, 89124 Reggio Calabria, Italy; giuseppe.micali@unirc.it (G.M.); antonio.mauceri87@unirc.it (A.M.); francesco.sunseri@unirc.it (F.S.); mrabenavoli@unirc.it (M.R.A.)
* Correspondence: antonio.tiberini@crea.gov.it
† These authors contributed equally to this work.

**Abstract:** The onion yellow dwarf virus (OYDV) represents a limiting biotic stress in onion (*Allium cepa* L.); little information is available regarding resistant varieties. In Italy, onion production is limited but represented by a wide diversity of ecotypes. A two-year trial was carried out to test the OYDV-susceptibility/tolerance of different Italian onion cultivars by rating symptom severity and plant growth parameters and assessing post-harvest secondary infections. The cultivar and ecotypes included in the study were characterized by simple sequence repeats (SSR) analysis, and the expression analysis of two genes (Eukaryotic translation initiation factors, *EIFs*) involved in potyvirus replication was also performed. Two susceptible and one tolerant cultivar were identified based on symptom expression and virus impact on plants. Although differences in growth parameters were limited to the first-year trial, the infection was correlated to a higher incidence of secondary infections in post-harvest, with altered water balance in infected bulbs. This correlation was also demonstrated during the long-term storage of bulbs. SSR analysis identified different clusters and only one gene isoform (*EIF4e*iso1) showed different expression levels in the OYDV/onion pathosystem. In conclusion, this study defines the genetic profile of Italian onion cultivars and provides evidence on susceptibility/tolerance features which will be useful in the future for the identification of viral resistance traits in onion.

**Keywords:** onion yellow dwarf virus; onion cultivars; EIF genes; SSR analysis; post-harvest infections





## 1. Introduction

Onion (*Allium cepa* L.) is the most cultivated species of the genus *Allium* (family *Liliaceae*) with a production of about 4500 k tons/year and 215 k hectares cultivated worldwide (FAOSTAT, 2021, https://www.fao.org/faostat/en/#data/QCL, accessed on 23 April 2023). Onion cultivation needs limited climatic requirements, being adapted to both tropical and temperate conditions. Since ancient times, onion has been an important vegetable crop for people around the world, used as fresh, cooked, and dried food, but also for religious and medical practices. Nowadays, it is assumed also as a functional food due to its abundance in bioactive antioxidant compounds, such as polyphenols and flavonoids [1].

Many diseases and disorders can affect onion crop. Among these, the EPPO Database reports two viruses belonging to the genus *Potyvirus* (family *Potyviridae*): onion dwarf yellow virus (OYDV) and leek yellow stripe virus (LYSV) [2]. OYDV is reported as the most widespread potyvirus affecting onion and other *Allium* spp. and causes harmful effects on crop yield and bulb quality. The virus, firstly reported in the United States (USA) in 1929 [3],

was found in Italy in 1983 [4]. In Southern Italy, a high incidence of OYVD-infection was observed in the onion cultivar 'Rossa di Tropea' [5], representing one of the production damages caused in this area. OYDV-infection induces longitudinal chlorosis along the leaves, dwarfism, leaf flattening, crinkling, and yellowing symptoms, that determine bulb size and weight reduction up to 40% [6,7]. Other frequent symptoms, such as distortion and curling of flower stems, cause a lack of seed production up to 50% [6]

OYVD is transmitted, in a non-persistent manner, by more than 50 aphid species [8], including *Myzus persicae* (Sulzer), *Aphis craccivora* (Koch), and *Aphis gossypii* (Glover) [9], and one kilometer of distance between seedbeds and OYDV infected onion crops is suggested to guarantee OYVD-uninfected bulb cultivation [10]. OYDV-infection remains persistent in onion sprouts, plant residues, and bulbs. Virus transmission by seed was observed only in a few Egyptian cultivars [10–12]. The management strategies of virus infection are limited to vectors control and the use of healthy germplasm and/or onion cultivars resistant/tolerant to OYDV.

In Italy, onion production is characterized by a high variable germplasm, selected over time for bulb morphological differences and organoleptic qualities. These traits led us to define different ecotypes that are sometimes related to very restricted cultivation areas. Little information is available about the resistance of these onion ecotypes to OYDV, and experimental trials on their responses to the infection have not been reported so far. Moreover, the abundance of germplasm in Italy is still poorly described at the phylogenetic level, so it is important to investigate the genetic characterization of these cultivars.

Cultivated and wild plant species co-evolved with potyviruses, resulting in the development of recessive resistance in plants and/or adapted virulent isolates. In both cases, the mechanism involves the Eukaryotic translation initiation factor (EIF) plant family genes and mainly two gene isoforms *EIF4e* and *EIF4g* associated in the *EIF4f* complex that show a key role in potyvirus infection [13–15]. Since viruses do not possess their own translation factors, they employ different strategies to synthetize proteins, such as recruiting host *EIF genes*, through the modulation of *EIF4e* and *EIF4g* activity in favour of viral, rather than plant, protein synthesis [14]. In other potyvirus-plant pathosystems, it had been shown that *EIF4e* or *EIF4g* mutations can confer resistance to plant viruses [16–18], but little information is available in the frame of OYDV-onion interaction, despite the *EIF* gene modulation that could compromise the growth and development of OYDV-infected onion plants [14,19], causing alterations in the onion bulbs physiology, such as changes in water balance and accumulation in leaves and bulbs [20].

In this study, experimental trials to identify tolerant and susceptible Italian cultivars/ecotypes, selecting the most valuable in terms of organoleptic properties, were carried out; the selection was also made to include different geographical origins in order to enhance genetic variability and maximise the possibility of finding tolerant traits. Phylogenetic studies were conducted by simple sequence repeats (SSR) analysis to identify the genetic distance among the cultivars and to correlate their susceptibility to OYVD to shared genetic profiles. In addition, the functionality of EIF genes involved in OYDV pathogenicity was analyzed in OYDV-tolerant and -susceptible cultivars by a multiple-sequence alignment of conserved motifs and quantitative relative expression analyses.

## 2. Materials and Methods

### 2.1. Plant Material and OYDV Inoculation

A total of 25 accessions of onion, including 14 different cultivars, of which one represented by 12 ecotypes (Acquaviva 1–12), were retrieved from cooperatives, European and Italian germplasm collections, and nurseries. The only OYDV cultivar actually reported as resistant was included as the control (Texas early grano 502) (Table 1).

**Table 1.** Accessions of onion germplasm used in the trial. * Acquaviva 1–12 were twelve different ecotypes of the Acquaviva cultivar.

| Accessions | Germplasm Collection | Origin |
|---|---|---|
| Rossa di Tropea (Reggio Calabria-RC) | University of Reggio Calabria | Calabria |
| Giarratana | Private nursery | Sicilia |
| Bianca di Castrovillari | ARSAC–Castrovillari (Cosenza-CS) | Calabria |
| Pera Sanguigna di Peschici | University of Foggia | Puglia |
| Rossa Margherita | University of Foggia | Puglia |
| Bianca di Margherita | University of Foggia | Puglia |
| Rosa o Dorata di Monteleone | University of Foggia | Puglia |
| Acquaviva (1–12 *) | University of Bari | Puglia |
| Ramata di Montoro | Coop.Rama | Campania |
| Dorata d'Ozieri | Genebank information system of the IPK Gatersleben | Sardegna |
| Dorata di Voghera | University of Pavia | Lombardia |
| Rossa di Breme | University of Pavia | Lombardia |
| Paglierina di Sermide | University of Pavia | Lombardia |
| Texas early grano 502 | Private seller | Commercial cultivar |

Two trials were conducted in two consecutive years to assess the OYDV symptom severity as well as the impact of the virus on the physiology of plants and bulbs after a mechanical OYDV-inoculation. The first trial included all the selected accessions and the second one was set up after a cultivar selection based on the OYDV susceptibility/tolerance resulted from the first-year, in particular; 'Acquaviva 7' and 'Pera sanguigna di Peschici' as susceptible cultivars, 'Rossa di Tropea (Reggio Calabria–RC)' as tolerant cultivar, plus 'Rossa di Tropea (Vibo Valentia-VV)', another ecotype of the same cultivar, and 'Texas early grano 502' as resistant control cultivar.

The plants used in this study were grown inside an insect-proof greenhouse located at the Department of Agraria, University *Mediterranea* of Reggio Calabria (Italy), to preserve their phytosanitary status. The onion seeds were sowed for germination on a substrate composed of purple peat and perlite, then the 10 cm tall seedlings were transplanted (two plants per pot) and growth up to bulb formation. The 20 cm pots were previously filled with a mixture of sandy and clayey soil in a 3:1 ratio, suitable for growing bulbs.

In the first- and second-year experimental trial, 30 plants for each genotype were analyzed as follows: one month after transplantation, according to agronomic procedures (four months after sowing) 22 plants were mechanically OYVD-inoculated, leaving the remaining 8 plants as healthy controls. Mechanical inoculation was performed through 10 needle punctures along two different leaves for each plant to simulate the probing activity of aphid vectors. Before each puncture, the syringe needle was soaked in a sap of OYDV-infected leaves (OYDV-5 strain provided by the Department of Agraria, University *Mediterranea* of Reggio Calabria, previously characterized [21]), extracted in presence of phosphate buffer (0.1 M; 1:5 *w:v*). At 30 days post inoculation (dpi) both inoculated and not-inoculated plants were analyzed by DAS-ELISA [22] using a commercial kit (Bioreba Reinach, Reinach, Switzerland) and following the manufacturer's instructions. The assays were performed in duplicate, including the healthy and positive controls provided by the kit. The reactions were measured by spectrophotometer Multiscan FC (Thermo Scientific, Carlsbad, CA, USA) at 405 nm, and samples with an optical density double of the healthy control were considered positive.

## 2.2. Evaluation of Susceptibility/Tolerance to OYDV

### 2.2.1. Assessment of Symptoms Severity

After the mechanical OYDV-inoculation, the number of OYDV-infected plants was recorded according to DAS-ELISA results. At 30 days post inoculation (dpi), OYDV symptoms severity was evaluated using the following score: 0 = no symptoms, 1 = mild symptoms, 2 = mild-medium symptoms, 3 = medium symptoms, 4 = medium-severe symptoms, 5 = severe symptoms, according to the yellowing, twirling, and dwarfing intensity (one value for each symptom) (Supplementary Figure S1).

Plants resulted OYDV-positive by DAS-ELISA were divided by the number of plants inoculated (22) within each accession and reported as a percentage to calculate the infection rate (IR %). The sum of the score assigned to each symptom of all the infected plants within each accession was divided by the number of plants detected as OYDV-positive by DAS-ELISA to obtain the plant infection index (PII). This index, which was further reported as a percentage value, assuming the PII highest value as 100%, was applied to estimate the overall symptomatology severity of OYDV-infected plants for each accession.

### 2.2.2. Effect of ODYV on Plant Growth, Bulb Long-Term Storage and Bulb Water Losses

Plant growth parameters were evaluated by means of the longest leaf height (cm) and the number of leaves recorded in infected and healthy plants at 15 and 30 dpi. Then, all plants of the trial were removed at the almost complete bulbs growth (45 dpi) and left on the soil of the greenhouse until the leaves resulted completely dry (60 dpi). Afterward, the bulbs were stored in a warehouse for 60 days to evaluate the OYDV effects on the presence/absence of secondary pathogen infections and on the ability to maintain the water after storage. The bulb water loss between infected and uninfected bulb and among genotypes was analyzed by weighting 1 g from each bulb before and after its complete dehydration in oven at 60 °C. Percentage of weight reduction was obtained by the ratio: fresh weight-dry weight/fresh weight × 100. The interaction between the phytosanitary state of the plants and the plant growth parameters (height of the higher leaf and number of leaves produced) at 15 and 30 dpi in the first and second-year trial were evaluated by ANOVA with Microsoft® Excel® 2019 MSO (Version 2311 Build 16.0.17029.20108), using Tukey's test as a post hoc test to separate statistically significant means.

## 2.3. SSR Genetic Analysis

The genetic distance between the accessions was evaluated by SSR analysis using 12 onion SSR markers [23,24] (Table 2). Among the 'Acquaviva' ecotypes, only 'Acquaviva 7' was included for its higher susceptibility to OYVD.

**Table 2.** SSR markers used in onion cultivars genetic analysis.

| Primer | Forward | Reverse | Size | Dye |
|---|---|---|---|---|
| ACM 373 | AGGTTAAGAAGTTGAATGGTCTG | AAATGGACAAGTGGCATTCA | 145–159 | FAM |
| ACM 101 | CCTTTGCTAACCAAATCCGA | CTTGTTGAGAAGGAGGACGC | 227–248 | VIC |
| ACM 235 | TGAGTCGGCACTCACCTATG | ACGCATTTTCAAA TGAAGGC | 292/304 | PET |
| ACM 446 | TCAAGAATTCTGTTGCATCTTGT | AATAAGACCGCAGAAACGAAA | 122–124 | FAM |
| ACM 449 | GTAAAGGTGTAATAGGAATGAATCG | TACAAAGAAACACACGCGCT | 133–148 | VIC |
| ACM 045 | AAAACGAAGCAACAAACAAAA | CGACGAAGGTCATAAGTAGGC | 226–275 | NED |
| ACM 138 | ACGGTTTGATGCACAAGATG | CCAACCAACAGTTGATACTGC | 242–286 | NED |
| ACM 387 | ACGCACACTATTTGGGAAGG | GAGGAATAGAGAAGGCTGCG | 151–162 | PET |
| ACM 134 | ACACACACAAGAGGGAAGGG | CACACACCCACACACATCAA | 198–212 | FAM |
| ACM 119 | TTTCAGCAACATAGTATTGCGTC | TCTTCGGGATTGGTATGGAG | 241–259 | PET |
| ACM 443 | TGGTGCTTGCTATGTTTTGC | CCCTAGGCCAAGCTTACTTGT | 154–179 | NED |
| ACM 477 | TGCAATTGGAACTTTGGTTTT | CCGTTCCTCTATTTTGCAGC | 160–165 | VIC |

### 2.3.1. DNA Extraction and PCR Analysis

Total DNA was extracted from young leaves using the DNeasy Plant Mini Kit (Qiagen, Hilden, Germany) according to the supplier's protocol. PCR analysis was carried out in a final volume of 10 µL, containing 7–50 ng of genomic DNA, 2 µL of 5X GoTaq Buffer (Promega, Madison, WI, USA), 0.075 µmol of each forward and reverse primer, 0.2 mM dNTP (Invitrogen, Waltham, MA, USA) and 0.25 U of GoTaq G2 DNA Polymerase (Promega, Madison, WI, USA). The reaction was performed in a Mastercycler nexus gx2 thermocycler (Eppendorf, Hamburg, Germany), using the following thermal profile: initial denaturation at 95 °C for 2 min, followed by 40 cycles at 95 °C for 30 s, annealing 56 °C for 30 s, elongation 72 °C for 20 s, and a final extension at 72 °C for 5 min.

PCR products were separated on a capillary electrophoresis system (ABI PRISM 3500 Genetic Analyzer, Applied Biosystem, Waltham, MA, USA), using 1 µL amplicon, 0.5 µL of GeneScanTM 600 LIZ® Size Standard v2.0 (Applied Biosystems, Waltham, MA, USA) as internal control and 8.5 µL of Hi-DiTM Formamide (Applied Biosystems, Waltham, MA, USA).

### 2.3.2. Statistical Analysis

The data were analyzed using Gene Mapper v.5 software (Applied Biosystem, Waltham, MA, USA). Population analysis and descriptive statistics via microsatellite loci to estimate the genetic distance (GD) were performed using the GenAlex software version 6.5 [25], by which Na = Number of Different Alleles is the total number of different alleles present in a population for a given locus; Ne = Number of Effective Alleles is the number of alleles that actually contribute to the genetic diversity of a population, taking into account their relative frequency; I = Shannon's Information Index is a measure of the genetic diversity of a population, taking into account both the number of alleles present and their relative frequency; Ho = Observed Heterozygosity is the proportion of individuals in a population that are heterozygous for a given locus; Expected Heterozygosity is the proportion of individuals in a population that would be heterozygous for a given locus if the population were in Hardy–Weinberg equilibrium; PIC = Polymorphic Information Content is a measure of the ability of a genetic marker to distinguish between different individuals; F = Fixation Index is a measure of the deviation from Hardy–Weinberg equilibrium in a population. The genetic distance matrices and their genetic relationships obtained from the pairwise comparison and the group of populations were reported in a two-dimensional chart by Principal Coordinate Analysis (PCoA). A phylogenetic tree based on Nei's genetic distance coefficient [26] and the UPGMA (Unweighted Pair Group Method with Arithmetic Mean) algorithm was generated by Mega X software (version 10.1.8).

### 2.4. Conserved Motifs among EIF4e and EIF4g

A series of bioinformatics steps aimed to identify and analyze conserved motifs within *EIF4e* and *EIF4g* proteins across diverse plant species were performed, in the genomes of *Arabidopsis thaliana* (www.arabidopsis.org, accessed on 20 January 2023) and *Solanum lycopersicum* (https://solgenomics.net/, accessed on 20 January 2023). The retrieval of *EIF4e* and *EIF4g* protein sequences from both the *Arabidopsis thaliana* and *Solanum lycopersicum* genomes served as the foundational dataset for subsequent comparisons and motif discovery. The obtained sequences were then subjected to multiple-sequence alignment to pinpoint regions of sequence conservation or similarity.

Following alignment, a phylogenetic tree was constructed to visually represent the evolutionary relationships between *EIF4e* and *EIF4g* sequences from various plant species. This dendrogram aids in gaining insights into the evolutionary history and divergence of these proteins over time. The next step involves comparing the aligned *EIF* sequences to the Onion Transcriptome Database (OTD, http://onion.riceblast.snu.ac.kr/download.php, accessed on 28 January 2023) to uncover potential homologous relationships with onion genes as documented in the OTD.

To discern conserved motifs within *EIF4e* and *EIF4g* proteins and their orthologues, the MEME online tool, version 5.0.5, was employed. MEME is a robust resource for motif discovery, allowing to identify short, conserved sequences or patterns that may bear functional significance within the protein sequences [27].

### *2.5. Gene Expression Analysis of the Eukaryotic Translation Initiation Factors (EIF)*

### 2.5.1. Plant Material and RNA Extraction

Gene expression analysis of *EIF4e* and *EIF4g* was performed on leaf and bulb samples collected from the second-year trial. Total RNA (TRNA) was extracted from 0.2 g of leaf and bulbs tissues following the manufacturer's instruction of RNeasy Plant Mini Kit (Qiagen, Hilden, Germany). The purity and quantification of the TRNA was determined with Nanodrop 2000 (Thermo Scientific, Wilmington, NC, USA) and integrity was assessed through an electrophoretic run on 1% *w/v* agarose gel.

### 2.5.2. Primers Design and Housekeeping Gene Detection

The Basic Local Alignment Search Tool (BLAST, https://blast.ncbi.nlm.nih.gov/Blast.cgi, accessed on 30 January 2023) was employed to discover regions of local similarity between homologous sequences of the *EIF4e* and *EIF4g* genes. *Asparagus officinalis* was used as the reference organism to perform sequences comparison (species closest to the onion among the available); the *EIF4e* and *EIF4g* sequences from Genbank (https://www.ncbi.nlm.nih.gov/genbank, accessed on 30 January 2023) with identifier XP_020246192.1 and XP_020240923.1, respectively, were used as input sequence to obtain homologous sequences in onion by the OTD (Onion Transcriptome Database) (http://onion.riceblast.snu.ac.kr/download.php, accessed on 28 January 2023). Primer design for the EIF4 isoform was performed using PRIMER3 (http://primer3.ut.ee/ accessed on 30 January 2023) to design the primers (Table 3).

**Table 3.** Primers of *EIF* and *actin* gene used for gene expression analysis.

| Primer | Forward | Reverse | Ta (°C) | Conc. (μmol) |
|---|---|---|---|---|
| *EIF4e* iso 1 | GAGGACCCTGTTTGTGCCA | GTGTGCATTTTTGGTCCAT | 59 | 0.5 |
| *EIF4e* iso 2 | GTCACCCAAGCAATTTAATG | GATTTTCCTCG TGAACAGTTGAC | 59 | 0.6 |
| *EIF4g* iso 1 | CCCTCAGTGTTGCCTTCTCC | TTCCATATCCACGCTGAGGC | 59 | 0.2 |
| *EIF4g* iso 2 | TGGCAGGAGAGAAGGAAGGT | GGTCCTCGTGTCAGTCTGTT | 59 | 0.5 |
| Actin | CTGGGATGACATGGAGAAGATT | GTTAAGTGGAGCCTCCGT | 59 | 0.1 |

The onion *actin* gene was used as the reference housekeeping gene [28]. The specificity of each primer was confirmed by evaluating the size of the expected PCR product.

### 2.5.3. Reverse Transcription and Gene Expression Analysis by qPCR

The reverse transcriptase reaction was performed following the manufacturer's instructions of the Maxima First Strand cDNA Synthesis Kit for RT-qPCR, with dsDNase (Thermo Fisher Scientific, Waltham, MA, USA) adding 200 ng of TRNA in a final volume of 20 μL. The product of this step was diluted 1:10 and an aliquot of 1 μL was added in 10 μL final volume reaction with 5 μL of PowerUP SYBR Green Master Mix (Applied Biosystem, Waltham, MA, USA) and 0.1–0.6 μmol of forward and reverse primers (Table 3). The reaction was performed in a StepOneTM Real-Time PCR System (Applied Biosystems, Waltham, MA, USA) thermocycler, setting the following thermal profile: 50 °C for 2 min, followed by a step at 95 °C for 2 min, and 40 cycles of: a denaturation phase for 15 s at 95 °C, an annealing/elongation step for 60 s at 59 °C.

The analysis was carried out on three biological replicates and in three technical replicates. A relative standard curve for each gene was generated using a triple serial dilution of cDNA pools (obtained by mixing the same proportion of all cDNA samples)

using StepOne software v2.3 (Applied Biosystems, Waltham, MA, USA). The PCR efficiency of the primer pairs has been optimized to be in the range 92–100% with R2 values of 0.996. In addition, amplification specificity was assessed by melting curves for the presence of a single peak.

The calculation of gene expression was based on the $2^{-\Delta\Delta Ct}$ method [29] using actin gene as the reference gene. Data of relative gene expression were processed by two-way ANOVA with Minitab Statistical Software Version 21.1.0, based on "cultivar" and "condition" (infected/not-infected) as main factors, and Tukey's honest significant difference (HSD) ($p < 0.05$) test.

## 3. Results

### 3.1. Evaluation of Susceptibility/Tolerance to OYDV

3.1.1. Assessment of Symptoms Severity

In the first-year trial, OYDV-inoculated plants showed an average percentage of infection rate of 58%. The PII (%) clearly showed that 'Rossa di Tropea' and 'Texas early grano 502' were the most OYDV-tolerant cultivar (PII (%) = 1.3 and 6.7, respectively); 'Acquaviva 7' was the most susceptible (PII (%) = 100), followed by 'Pera Sanguigna di Peschici' (PII (%) = 97.9). The data from the second-year trial, performed using a total of four cultivars selected by means of the most tolerant/sensitive cultivar, with the addition of the 'Rossa di Tropea' (VV) ecotype, confirmed the differences in tolerance/susceptibility resulted from the first-year trial. 'Rossa di Tropea' (VV) was included as to double check and confirm the tolerance traits observed in this cultivar, reported to be susceptible to OYDV, indeed, this ecotype resulted to be the most OYDV-tolerant ecotype with a PII (%) value of 1.3. 'Pera Sanguigna di Peschici' showed the highest PII (%) value (100) for this trial (Table 4).

**Table 4.** Results of first and second trial symptom severity observations and DAS-ELISA analysis at 30 dpi accomplished in susceptibility/tolerance evaluation trial. Genotypes with the star were chosen for the second-year trial, together with an extra ecotype of 'Rossa di Tropea' was added: 'Rossa di Tropea (VV)'. In asterisks indicate the accessions further included in the second-year trial.

| First trial Accessions | Total Symptoms Score | OYDV-Positive Plants (n) | Infection Rating (IR %) | Plant Infection Index (PII) | PII (%) § |
|---|---|---|---|---|---|
| Acquaviva 1 | 46 | 11 | 50 | 4.18 | 43.9 |
| Acquaviva 2 | 89 | 12 | 55 | 7.42 | 77.9 |
| Acquaviva 3 | 70 | 14 | 64 | 5 | 52.5 |
| Acquaviva 4 | 107 | 14 | 64 | 7.64 | 80.2 |
| Acquaviva 5 | 104 | 16 | 73 | 6.5 | 68.2 |
| Acquaviva 6 | 55 | 14 | 64 | 3.93 | 41.2 |
| Acquaviva 7 * | 143 | 15 | 68 | 9.53 | 100 |
| Acquaviva 8 | 88 | 14 | 64 | 6.29 | 66 |
| Acquaviva 9 | 66 | 12 | 55 | 5.5 | 57.7 |
| Acquaviva 10 | 78 | 14 | 64 | 5.57 | 58.5 |
| Acquaviva 11 | 135 | 16 | 73 | 8.44 | 88.5 |
| Acquaviva 12 | 72 | 13 | 59 | 5.54 | 58.1 |
| Rosa o Dorata di Monteleone | 50 | 16 | 73 | 3.13 | 32.8 |
| Bianca di Margherita | 20 | 6 | 27 | 3.33 | 35 |
| Pera Sanguigna di Peschici * | 112 | 12 | 55 | 9.33 | 97.9 |
| Bianca di Castrovillari | 74 | 14 | 64 | 5.29 | 55.5 |
| Rossa Margherita | 99 | 13 | 59 | 7.62 | 79.9 |
| Dorata d'Ozieri | 99 | 17 | 77 | 5.82 | 61.1 |
| Giarratana | 44 | 9 | 41 | 4.89 | 51.3 |
| Rossa di Tropea (RC) * | 1 | 8 | 36 | 0.13 | 1.3 |
| Ramata di Montoro | 74 | 13 | 59 | 5.69 | 59.7 |
| Texas early grano 502 * | 8 | 13 | 59 | 0.62 | 6.7 |
| Dorata di Voghera | 90 | 13 | 59 | 6.92 | 72.6 |
| Rossa di Breme | 38 | 8 | 36 | 4.75 | 49.8 |
| Paglierina di Sermide | 68 | 14 | 64 | 4.86 | 51 |

**Table 4.** *Cont.*

| Second trial accessions | | | | | |
|---|---|---|---|---|---|
| Acquaviva 7 | 22 | 11 | 50 | 2 | 56 |
| Pera Sanguigna di Peschici | 50 | 14 | 64 | 3.57 | 100 |
| Rossa di Tropea (RC) | 1 | 9 | 41 | 0.11 | 3.1 |
| Texas early grano 502 | 0 | 6 | 27 | 0 | 0 |
| Rossa di Tropea (VV) | 1 | 8 | 36 | 0.13 | 3.5 |

§ PII (%) = PII value × 100/trial highest PII.

### 3.1.2. Effect of OYDV on Plant Growth

The effect of virus infection on both leaf height and number observed during the first-year trial was assessed considering the cultivar both separately (Supplementary Figure S2) and as a whole (Figure 1a). In terms of the effect on the whole cultivars, it is possible to observe that the mean value of the highest leaf measured in OYDV-infected plants was lower (54.32 ± 10.34 SD) than in healthy plant (58.44 ± 9.54 SD) at 15 dpi ($p$ value < 0.001); this value was confirmed at 30 dpi (45.05 ± 10.21 SD in infected and 54.89 ± 9.62 SD in healthy plants; $p$ value < 0.001) (Figure 1a). In details, the values observed in the cultivar-by-cultivar analysis showed statistical differences for most part of the cultivars, especially at 30 dpi (Supplementary Figure S2a). The mean number of leaves produced by OYDV-infected plant showed no statistical differences (Figure 1a) compared to heathy ones ($p$ value > 0.05), both at 15 (6.36 ± 1.14 SD in infected and 6.67 ± 1.18 SD in healthy plants) and 30 dpi (5.9 ± 1.31 SD in infected and 5.87 ± 1.41 SD in healthy plants); the result was confirmed also analyzing cultivars separately (Supplementary Figure S2b).

The effect of virus infection on the plant growth was evaluted for the second-year trial, considering the cultivars/ecotypes both separately (Supplementary Figure S3) and as a whole (Figure 1b), as for the first-year trial; the evaluation was limited to the cultivars selected from the first year trial. As an overall result, the effect of OYDV-infection on both plant growth parameters was not significant ($p$ value > 0.05) at both 15 and 30 dpi. In particular, the height of the longest leaf in OYDV-infected plants had a mean value of 46.99 ± 10.87 SD and in healthy ones was 42.88 ± 10.96 SD, at 15 dpi; whereas at 30 dpi the mean value of infected plants was 50.49 ± 14.76 SD and 50.34 ± 17.22 SD in healthy ones, at 30 dpi. The number of the leaves in OYDV-infected plants was 6 ± 1.59 SD and 5.66 ± 1.78 SD in healthy ones at 15 dpi; the values of the second measurement at 30 dpi was 6.23 ± 2.39 SD in OYDV-infected plants and 6.26 ± 2.50 SD in healthy ones.

### 3.1.3. Effect of OYDV on Bulb Long-Term Storage and Water Loss

Visual inspection of the bulbs after a long-term storage highlighted a reduced tissue consistency of bulbs collected from the infected plants. In the first trial, internal rots had a higher incidence in the infected bulbs (62.50% in 'Acquaviva 7' and 14.3% in 'Pera sanguigna di Peschici') than in the healthy controls (0% in 'Acquaviva 7', 'Pera Sanguigna di Peschici', 'Rossa di Tropea (RC)' and 'Texas early grano 502') (Figure 2a). These observations were confirmed in the second trial in which a secondary infection was found in all the OYDV-infected plants belonging to the cultivar 'Acquaviva 7' (100%), 71% in plants of 'Pera sanguigna di Peschici' cultivar, 37.0% in plants of 'Rossa di Tropea (VV)' cultivar and 0% in plants of 'Rossa di Tropea (RC)' and 'Texas early grano 502' (Figure 2a); in healthy plants the incidence of secondary infections was 12.5% in plants of 'Acquaviva 7', 'Rossa di Tropea (RC)' and 'Texas early grano 502' cultivars, 14.3% in plants of 'Rossa di Tropea (VV)' cultivar and 0% in plants of 'Pera sanguigna di Peschici' cultivar (Figure 2b). The results suggest that OYDV-infection determines favorable conditions to rot development caused by post-harvest pathogens.

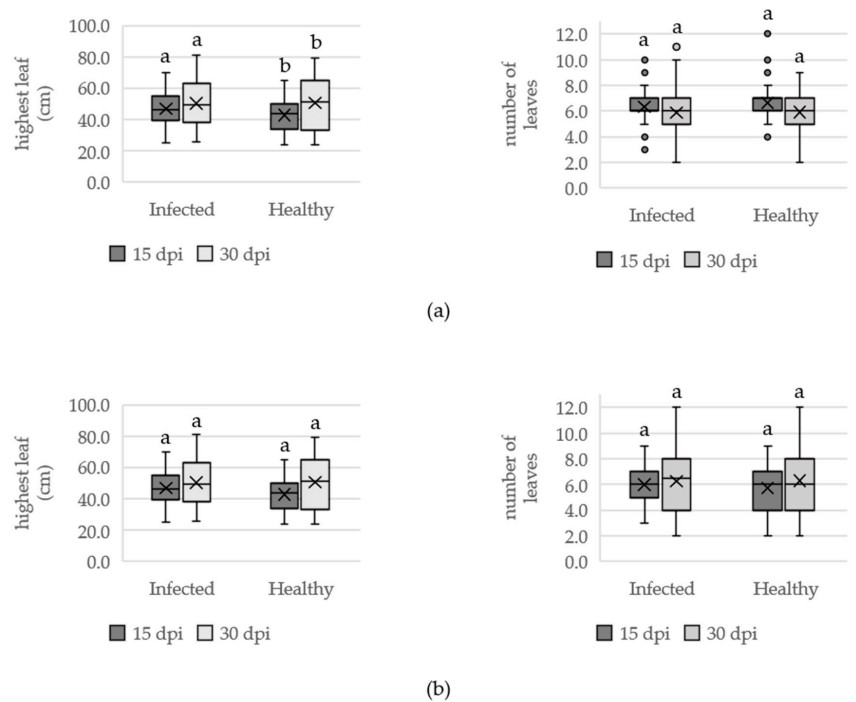

(a)

(b)

**Figure 1.** Effect of OYDV at 15 and 30 dpi on the highest leaf (cm) and number of leaves in the first- (**a**) and second-year trial (**b**). The results are expressed by mean ± the standard deviation and the X indicates the median value. Tukey's test as a post hoc test was used to separate means: different letters indicate statistically significant differences at *p* value < 0.05 among infected and healthy plants at 15 and 30 dpi.

Secondary infections

(**a**)

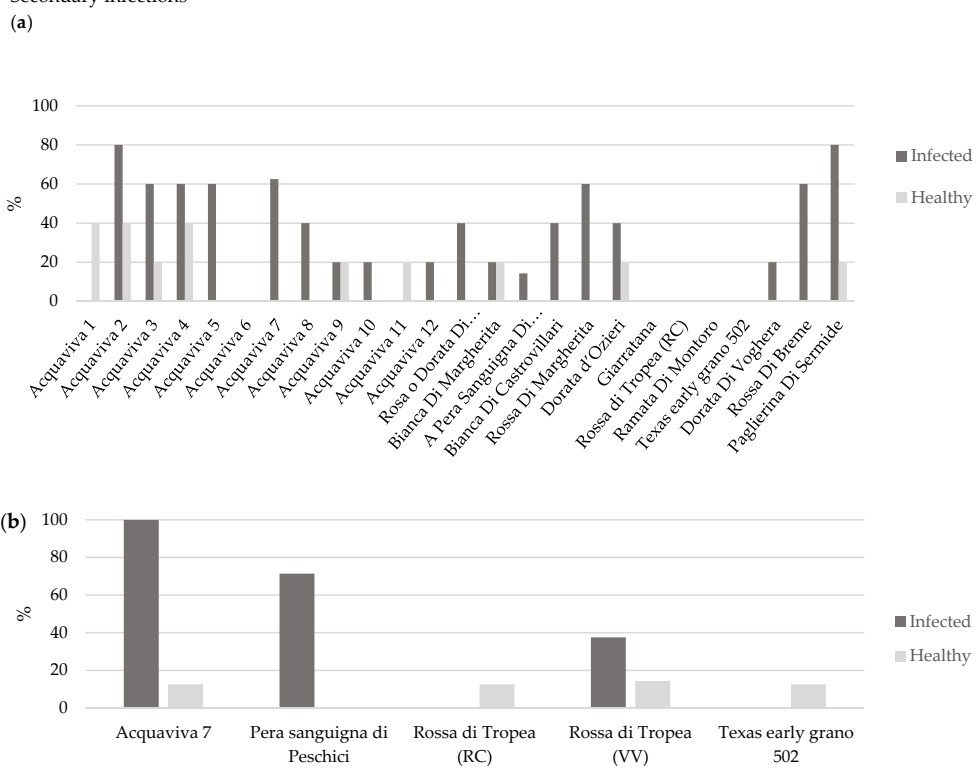

**Figure 2.** Percentage (%) of bulbs affected by secondary pathogens observed from infected and healthy plants belonging of each cultivar, detected by visual inspection during the storage phase in the first (**a**) and second (**b**) trial.

To investigate the possible correlation between a higher susceptibility to rots in OYDV-infected tissues and water content, the percentage index of water loss (percentage of weight reduction) was determined in bulbs of each cultivar after a long-term storage. 'Texas early grano 502' showed small differences in water content between healthy and infected bulbs (−0.17%), resulting in a stable water loss balance, regardless of the OYDV infection. Moreover, the tolerant genotype 'Rossa di Tropea' did not show a water content reduction (+1.36%). the susceptible cultivars 'Acquaviva 7' and 'Pera Sanguigna di Peschici', OYDV-infected, showed an increment of water loss content of +3.39% and +2.83%, respectively (Figure 3a). The differences in water loss content between susceptible and tolerant cultivars obtained in the second-year trial were: +1% in 'Acquaviva 7', +0.65% in 'Pera sanguigna di Peschici', −0.37% in 'Rossa di Tropea (RC)', −0.07% in 'Rossa di Tropea (VV)' and −0.06% in 'Texas early grano 502' cultivar (Figure 3b).

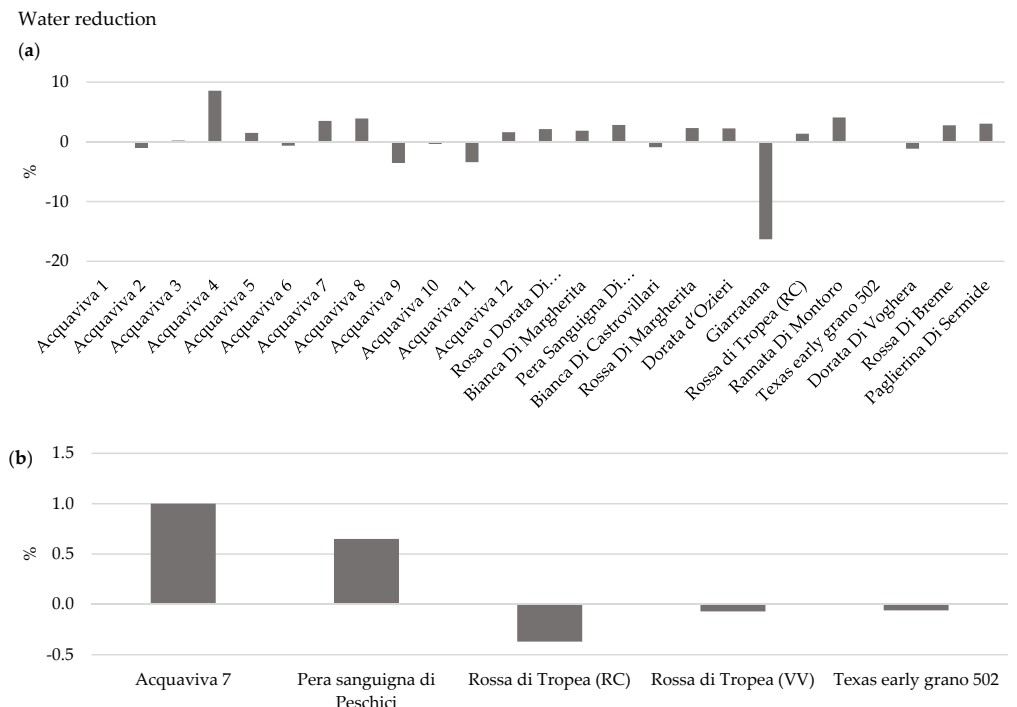

**Figure 3.** Difference in water reduction (%) between infected and heathy bulbs for each cultivar in first (**a**) and second (**b**) trial.

*3.2. SSR Genetic Analysis*

Eighteen onion cultivars/ecotypes belonging to different geographical areas were genotyped by selected SSR. Genetic data from 12 polymorphic SSR loci were analyzed by using Genemapper v.5, defining the length of each amplified fragment (Supplementary Table S1). SSR analysis of the onion collection are reported in Table 5 together with statistics and genetic summary.

The SSR markers amplified 99 alleles (Na) and a mean of 9.25 alleles per locus were detected. The number of alleles at each locus ranged from 5 (ACM235 and ACM446) to 13 (ACM373). The number of effective alleles (Ne) ranged from 2.980 (ACM235) to 9.184 (ACM373) with a mean of 4.91 alleles per locus. Shannon's Information Index (I) ranged from 1.284 (ACM235) to 2.378 (ACM373) with an average of 1.753. The observed heterozygosity (Ho) ranged from 0.2 (ACM119) to 0.867 (ACM138) with a mean of 0.662. The analyzed samples were at the Hardy–Weinberg equilibrium and the bias in the ratio between observed heterozygosity (H0) and expected heterozygosity (He) was not significant.

Indeed, the expected heterozygosity (He) ranged from 0.687 (ACM235) to 0.922 (ACM373) with a mean of 0.804. Polymorphic information content (PIC), ranged from 0.618

(ACM235) to 0.882 (ACM373) with a mean of 0.749. The fixation index (F) ranged from −0.026 (ACM138) to 0.900 (ACM235) with a mean of 0.437 (Table 5).

**Table 5.** Genetic diversity parameters among onion genotypes based on 12 SSR markers.

| Locus | Na [1] | Ne [2] | I [3] | Ho [4] | He [5] | PIC [6] | F [7] |
|---|---|---|---|---|---|---|---|
| ACM373 | 13 | 9.184 | 2.378 | 0.533 | 0.922 | 0.882 | 0.401 |
| ACM101 | 7 | 3.409 | 1.455 | 0.667 | 0.731 | 0.659 | 0.057 |
| ACM235 | 5 | 2.980 | 1.284 | 0.067 | 0.687 | 0.618 | 0.900 |
| ACM446 | 5 | 3.913 | 1.460 | 0.800 | 0.770 | 0.701 | −0.075 |
| ACM449 | 7 | 4.245 | 1.620 | 0.800 | 0.791 | 0.731 | −0.047 |
| ACM045 | 7 | 4.412 | 1.681 | 0.733 | 0.800 | 0.743 | 0.052 |
| ACM138 | 12 | 6.429 | 2.115 | 0.867 | 0.874 | 0.827 | −0.026 |
| ACM387 | 10 | 4.945 | 1.871 | 0.667 | 0.825 | 0.772 | 0.164 |
| ACM134 | 9 | 5.172 | 1.875 | 0.800 | 0.834 | 0.783 | 0.008 |
| ACM119 | 7 | 3.846 | 1.602 | 0.200 | 0.766 | 0.710 | 0.730 |
| ACM443 | 9 | 4.592 | 1.802 | 0.533 | 0.809 | 0.756 | 0.318 |
| ACM477 | 8 | 5.921 | 1.892 | 0.800 | 0.860 | 0.809 | 0.037 |
| Mean | 9.25 | 4.921 | 1.753 | 0.622 | 0.805 | 0.749 | 0.209 |

[1] Na = No. of Different Alleles; [2] Ne = No. of Effective Alleles; [3] I = Shannon's Information Index; [4] Ho = Observed Heterozygosity; [5] He = Expected Heterozygosity; [6] PIC = Polymorphic Information Content; [7] F = Fixation Index.

Cluster analysis based on these SSR data showed genetic relationships among the onion cultivars/ecotypes. The UPGMA tree showed demarcations among the cultivars/ecotypes included in the analysis, 'Rossa di Tropea' clustered separately, with a slight difference according to the different origin (Reggio Calabria-RC or Vibo Valentia-VV). Among the susceptible cultivars, 'Giarratana' and 'Bianca di Margherita' also formed a well-supported clade. 'Acquaviva 7' resulted to have the highest number of nucleotide changes, as well as 'Rossa di Margherita' (Figure 4).

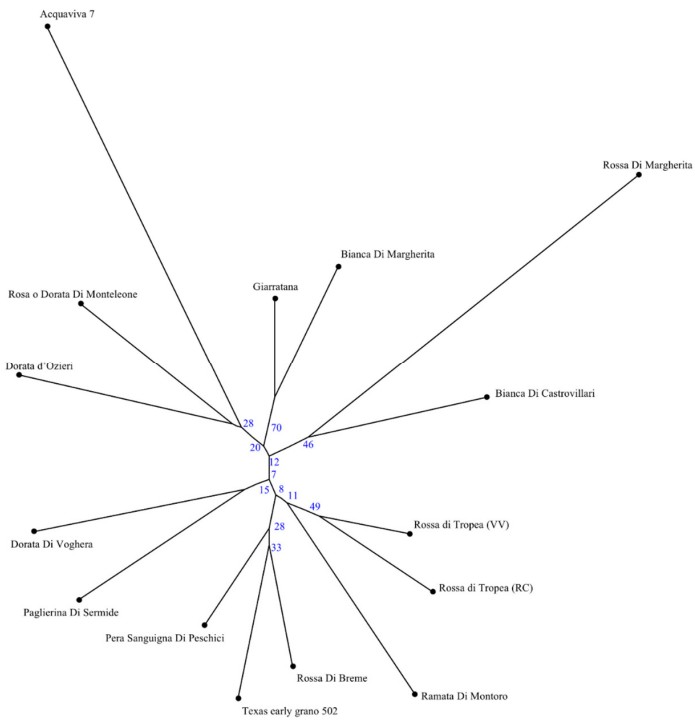

**Figure 4.** Genetic relationships among onion cultivars based on SSR data using Nei's genetic distance coefficient and the UPGMA algorithm on Mega X software (version 10.1.8). Bootstrap values about 70 are considered indicative of a robust phylogenetic rela-tionship, as in the case of Bianca di Margherita and Giarratana.

Principal Coordinate Analysis (PCoA) of 12 microsatellite loci is shown in a two-dimensional space (Figure 5). The scatter plot represents the sampled individuals, defined and labeled in terms of "Tolerance" and "Sensitivity" based on the criteria defined in the previous paragraphs. The total of genetic variance explained by PCoA is 26.4% (15.51% and 10.93% for PCoA1 and PCoA4, respectively), which indicates how much the principal coordinates explain the variability and therefore the differences in the population samples.

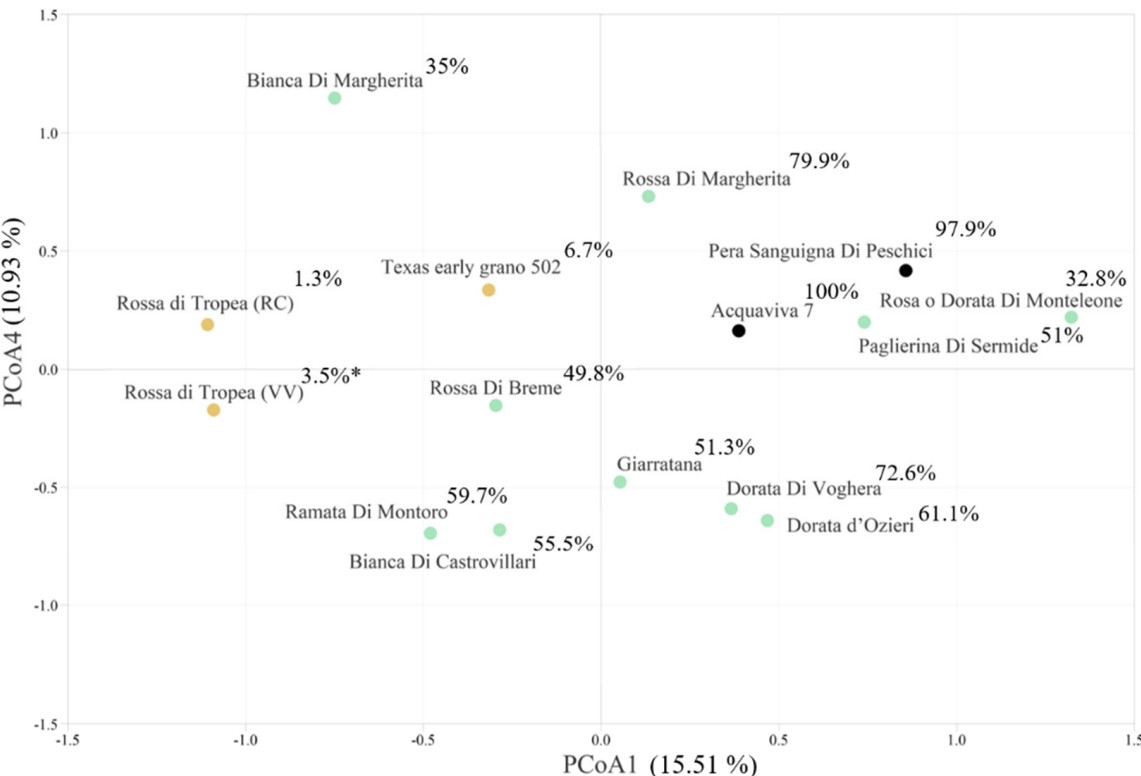

**Figure 5.** Principal coordinate analysis resulted from phylogenetic studies on the cultivars/ecotypes of the trial, considering of 12 microsatellite loci. In the graph, each cultivar/ecotype is represented by the PII (%) obtained in trials, with yellow and black dots representing the most tolerant and susceptible cultivar/ecotypes selected and included in the second-year trial. * PII (%) obtained in the second-year trial. Green dots represent the cultivars with intermediate levels of virus tolerability/susceptibility.

The PCoA of onion molecular markers shows a clear demarcation of assumed susceptible cultivars from the tolerant ones. 'Rosa o Dorata di Monteleone clustered close to 'Acquaviva 7' and 'Rossa Sanguigna di Peschici' despite showing a PII (%) value lower than 15%. 'Bianca di Margherita' assumes a clearer distance from the main cluster.

### 3.3. Conserved Motifs among EIF4e and EIF4g Interspecies

The phylogenetic analysis among *Allium cepa* and tomato *EIF4* isoforms, and all the *EIF4* family members from the arabidopsis genome highlighted similarity between tomato and arabidopsis sequences (Figure 6). In addition, to examine the structural organization of *EIF4* isoforms to identify the conserved domains the MEME online tool v5.0.5 was used (Figure 7). In the phylogenetic tree (Figure 6) all the isoforms of *EIF4g* gene (1 to 4) in onion clustered together along with tomato and arabidopsis sequences, as well as for the isoform 1 and 2 of *EIF4e.* No differences were observed in the conserved motifs in the onion isoforms of these two genes in comparison to tomato and arabidopsis sequences (Figure 7).

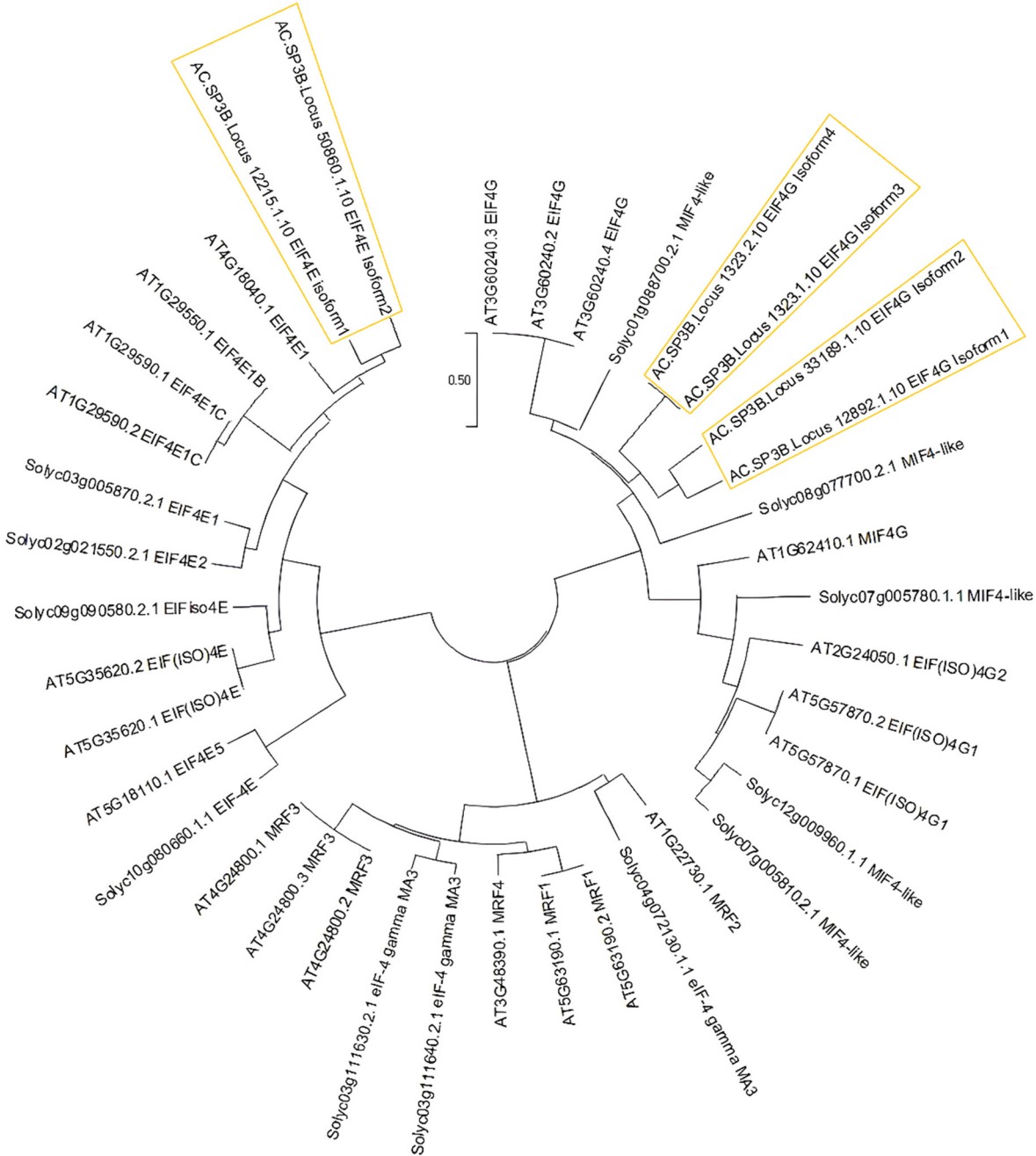

**Figure 6.** A phylogenetic tree of EIF4 peptide sequences from Arabidopsis (AT), tomato (Solyc) and onion (AC) (OTD, Onion Transcriptome Database) homologs. Samples used in our study are delineated by a yellow box. The sequences were aligned using Clustal*W* ClustalW in MEGA X software (version 10.1.8) and the phylogeny constructed using the neighbor-joining method.

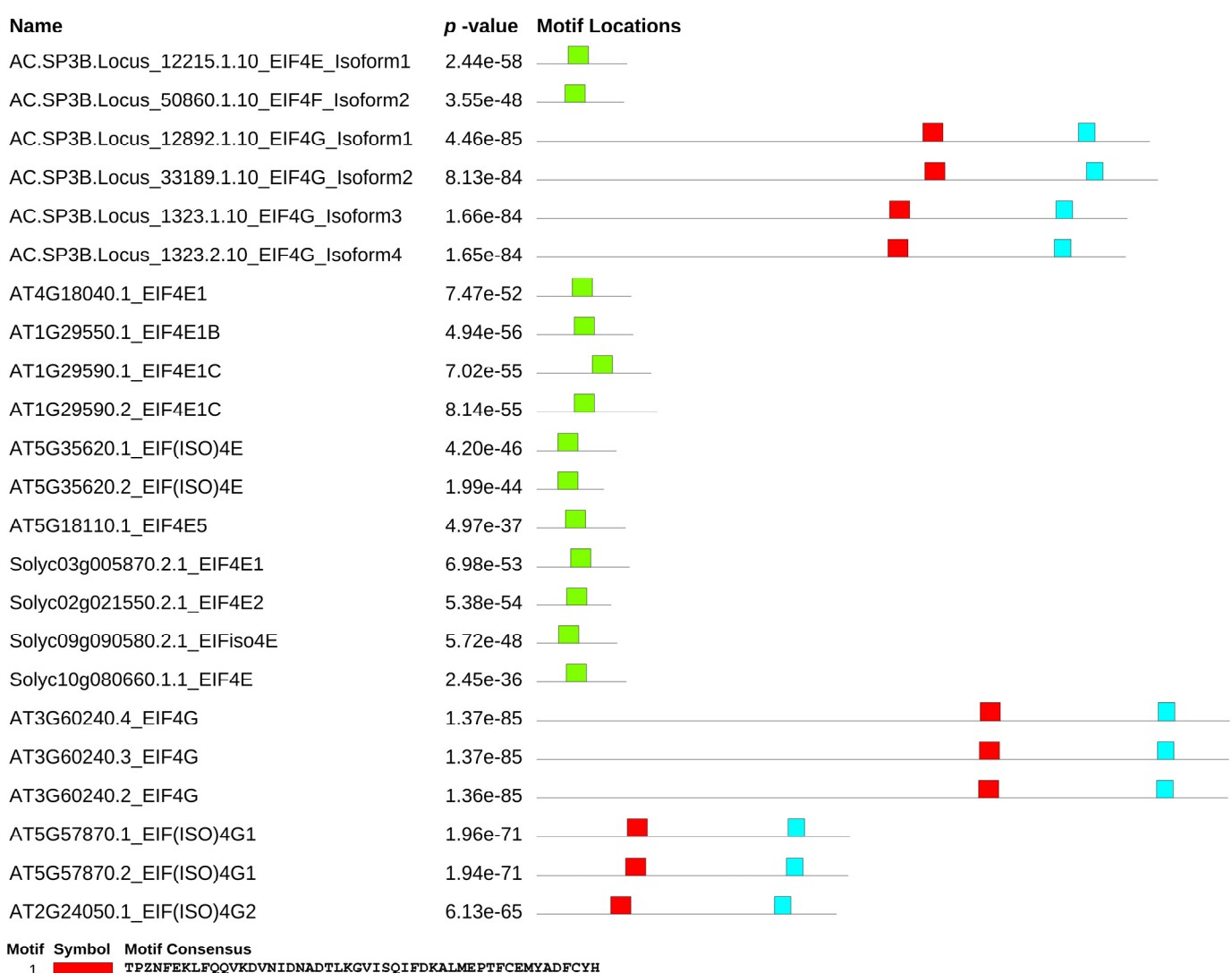

**Figure 7.** Analysis and comparison of sequence motifs in *EIF4* isoforms proteins from onion (AC), Arabidopsis (AT) and tomato (Solyc) by MEME online tool v5.05.

### 3.4. Gene Expression Analysis of the Eukaryotic Translation Initiation Factors (EIF)

Significant differences in transcript abundance were detected in leaf tissue (Figure 8) comparing four genotypes in the second trial ('Rossa di Tropea (VV)' was excluded). The *EIF4g* isoforms 3 and 4 were excluded because of the very limited levels of expression. In particular, *EIF4e* isoform 1 showed a variability affected by "condition" factor (health/infected, $p$ value < 0.007) indeed, the mean value of infected plants was about twice fold than the mean of healthy one. Variability of relative expression of *EIF4e* isoform 2 was determined by "cultivars" factor ($p$ value < 0.001), where the cultivars 'Texas early grano 502' and 'Pera sanguigna di Peschici' showed the major differences in transcript accumulation (Figure 8a). In the *EIF4g* isoform 1 the "cultivars" factor also explained a greater variance rate ($p$ value < 0.008), where 'Texas early grano 502' and 'Rossa di Tropea (RC)' showed a similar mean, higher than 'Pera sanguigna di Peschici'. The *EIF4g* iso 2 showed the lowest level of expression and significant variance for all the three factors "cultivars", "condition" and their interaction "cultivars*condition" was not detected (Figure 8b).

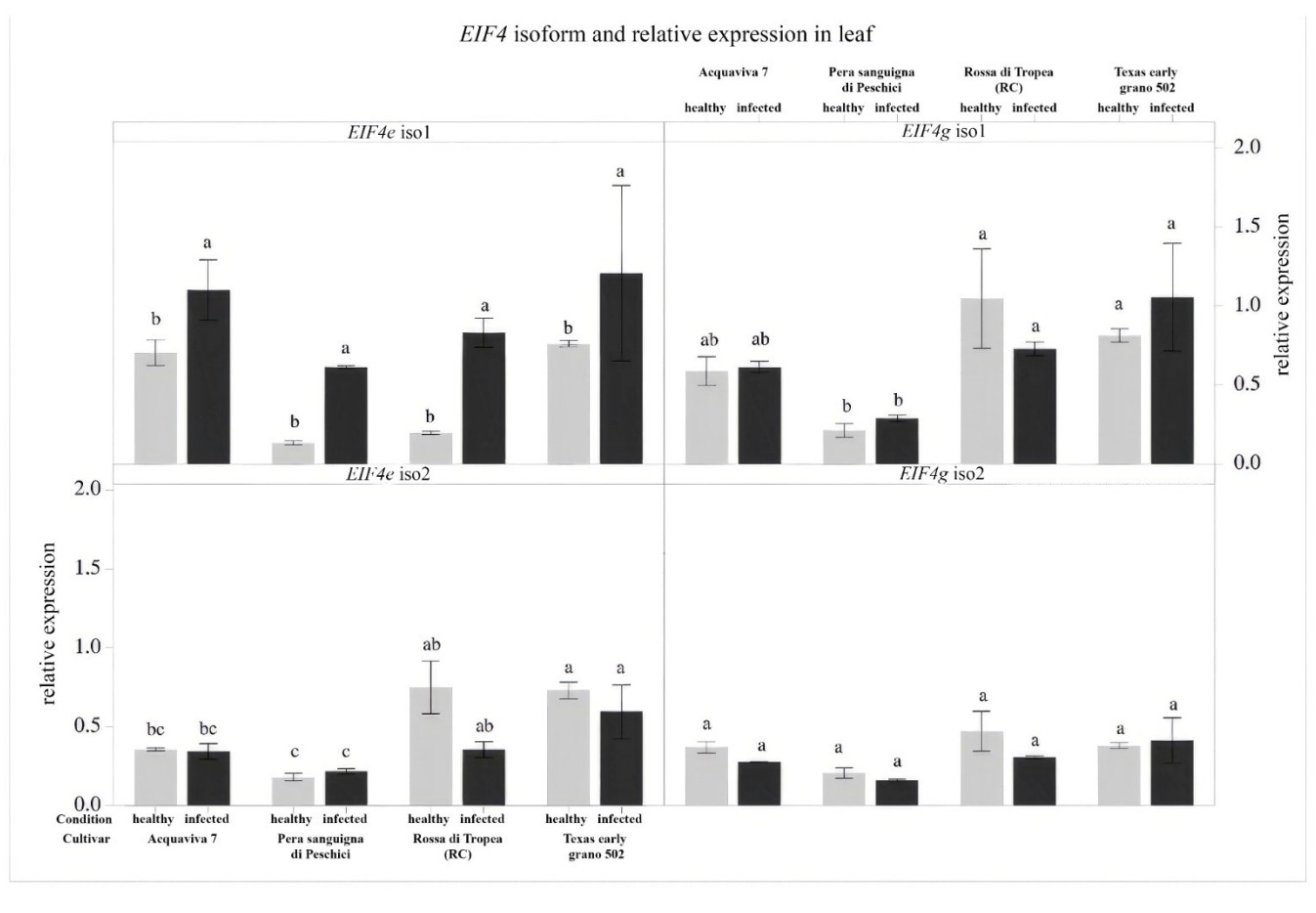

(**a**)　　　　　　　　　　　　　　　　　　　　　　　　　　(**b**)

**Figure 8.** Gene expression analysis of *EIF4e* (**a**) and *EIF4g* (**b**) isoforms in leaf tissue, expressed as the $2^{-\Delta\Delta Ct}$ method compared to the actin as reference gene). The results are expressed by mean ±. The standard deviation differences between means are determined using Tukey's honest significant difference (HSD) ($p < 0.05$): different letters indicate statistically significant differences between infected and healthy plants among different cultivars.

Instead, a significant different gene expression of *EIF4e* and *EIF4g* isoform was observed in bulbs tissues. In detail, the larger variability was explained by the interaction "condition*cultivars". The *EIF4e* isoform 1 did not show significant differences among the cultivars, the health condition, and their interaction ($p$ value > 0.05); by contrast, *EIF4e* isoform 2 showed different expression levels ($p$ value < 0.001) on 'Acquaviva' OYVD-infected bulbs compared to 'Texas early grano 502', 'Pera sanguigna di Peschici', 'Rossa di Tropea (RC)' OYVD-infected bulbs as well as not infected 'Texas early grano 502' (Figure 9a). The *EIF4g* isoform 1 showed a significant higher expression level ($p$ value < 0.06), in healthy 'Rossa di Tropea (RC)' bulbs compared to the other cultivars regardless OYDV infection. Finally, the *EIF4g* isoform 2 showed a level of gene expression not significantly different ($p$ value > 0.05) (Figure 9b).

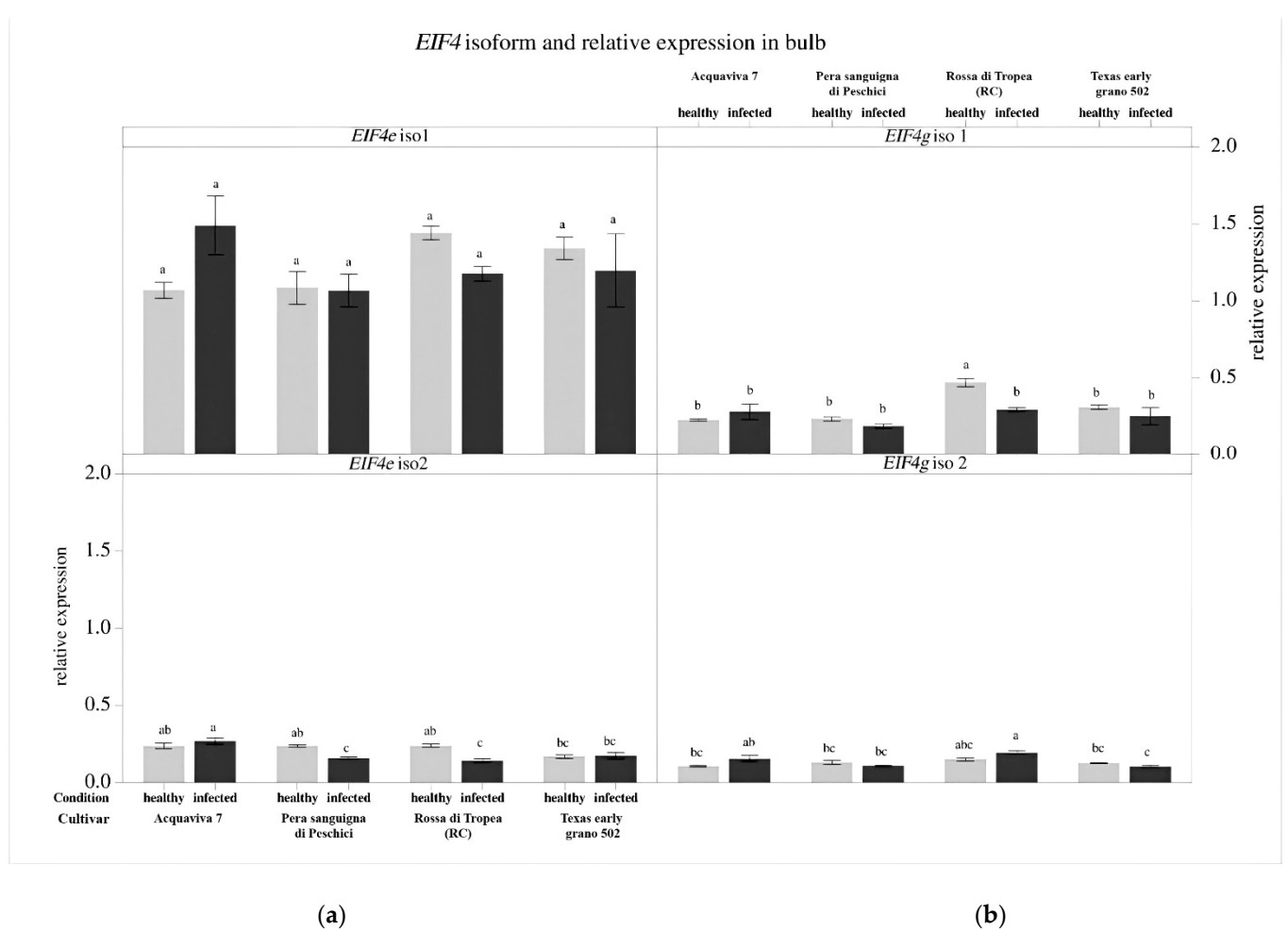

(**a**)

(**b**)

**Figure 9.** Gene expression analysis of *EIF4e* (**a**) and *EIF4g* (**b**) isoforms in bulbs tissue, expressed as $2^{-\Delta\Delta Ct}$ method compared to the actin as reference gene). The results are expressed by mean $\pm$ the standard deviation. Differences between means as determined using Tukey's honest significant difference (HSD) ($p < 0.05$): different letters indicate statistically significant differences between infected and healthy plants among different cultivars.

## 4. Discussion

Plant-virus interactions are a complex of processes derived by an unending battle between host defense response and pathogen antagonist action [30]. Viruses use host cell resources to support their own reproduction (replication) and dissemination (cell-to-cell and long-distance movements) interfering with the normal plant processes, and inducing physiological, biochemical, and structural changes and anomalies, often leading to a clear symptomatology. Most commonly, it is widely believed that virus infections are harmful to the host or havea negative acceptation, but little is known about the biology of plant viruses and their host in a natural system, regarding synergic or more complex interaction networking. Physiological and morphological responses to virus infection in plant depends by host species, virus–host recognition, virus species and/or strain [31]. OYDV is known to induce severe symptoms in *Allium* spp. including onion, as dwarfing, yellowing, stem twirling and losses in bulb production. Limited information is available about resistant/tolerant cultivars and how these react to OYDV infection, including the role of genes involved in pathogenicity.

Based on symptom inspection and PII%, 'Texas early grano 502' confirmed its OYDV-resistance along with 'Rossa di Tropea', whereas 'Acquaviva 7' and 'Pera Sanguigna di Peschici' were the most susceptible cultivars. OYDV appeared to determine an effect on the height of leaves in all the cultivars when considered as a whole, but not in all the single

cultivars when analyzed separately; in fact, this effect is not reported for some cultivars (Supplementary Figure S2a), among them the most tolerant cultivars ('Texas early grano 502' and 'Rossa di Tropea (RC)') were included, confirming that the infection does not affect the leaves developing in tolerance conditions. This effect is not confirmed in the second-year trial. By contrast, a correlation between occurrence of rots caused by secondary pathogens and susceptibility to OYDV was observed. Other studies regarding the OYDV-'Rossa di Tropea' pathosystem indicate that the virus infection could lead to a metabolism triggering of infected tissue, increasing the free water content and a consequential water loss effect [32]. In addition, a stomatal aberrant development in OYDV infected plants with consequent water content alteration was previously observed [33]. From the data obtained in this study it could be associated a direct effect of OYDV-infection to water accumulation; indeed, the most OYDV-susceptible cultivars showed an overall implement of water losses, but further studies have to be conducted to determine a statistical significance. 'Acquaviva 7' and the 'Pera Sanguigna di Peschici' cultivars showed the highest PII index, and a higher water content reduction compared to 'Rossa di Tropea' and 'Texas early grano 502' resistant cultivars. Our results suggest that OYDV-infection determine favorable conditions for rots development caused by post-harvest pathogens that could be associated with an altered water balance and a structural alteration of infected tissues [32]. Consequently, OYDV infection also has an impact on the bulb quality, affecting its shelf life, and this feature is reported as crucial for a correct long-term storage. The yield losses in this stage are reported ranging from 30 to 50% of the total product [34]. The OYDV infection control and a correct storage could help to prevent post harvesting pathogens infection [35]. Indeed, bulbs with low dry matter showed higher susceptibility to sprouting and post-harvest diseases [36].

Genetic analysis of eighteen onion cultivars/ecotypes from different geographical areas revealed significant genetic diversity, with a notable variation in the number of alleles and genetic diversity of the analyzed loci. Some onion cultivars, such as 'Rosa o Dorata di Monteleone', 'Acquaviva 7', 'Giarratana', 'Bianca di Margherita', and 'Bianca di Castrovillari', are genetically distinct from the others. Moreover, significant genetic differences have been observed between OYDV-tolerant onion cultivars such as 'Texas early Grano 502' and 'Rossa di Tropea' compared to the susceptible ones. These findings emphasize the importance of genetic diversity in the conservation and selection of onion cultivars/ecotypes, providing the basis for crop adaptation, disease protection, and species evolution over time [37]. Significant genetic diversity was detected among cultivars/ecotypes and distinct clusters emerged from clustering and PCoA, confirming the genetic differences between OYDV-tolerant and susceptible cultivars. These results underline the necessity to test a large germplasm collection for OYDV-susceptibility to identify resistant/tolerant genotypes useful to protect the onion productions [38]. *EIF4e* and *EIF4g* genes regulation is reported to have a crucial role in plant host-interaction, this mechanism is particularly important in virus infection [39,40], due to their ability to employ the translation initiation complex of the host to facilitate their own replication and translation of viral RNA [14]. Indeed, *EIF* genes represent a molecular target to quantify the severity of the infection in viral infections and phylogenetic studies on conserved motifs of this genes are useful to study the coevolution of the virus and its host. In particular, from the gene expression data obtained makes it clear that the *EIF4e* isoform 1 is strictly connected to the virus replication in leaves, confirming the crucial role of this gene in the virus replication for species belonging to potyvirus genus, whereas the other *EIF4e* isoform as well the *EIF4g* isoforms, seem to not represent a pathway differentially triggered by OYDV infection in onion. It is also clear that these genes were generally expressed in bulbs less than in leaves, despite the impact of OYDV infection on those tissues, suggesting an alternative pathway for the virus replication. This study highlighted that in onion, *EIF4e* isoform 1 represents a potential target for inducing mutations aimed at conferring resistance to OYDV, despite the molecular determinant related to the tolerance feature expression was not yet identified, requiring further studies. OYDV-infection exhibited a larger impact on *EIF4e* isoform 1 e expression in bulbs among different onion cultivars and it affected gene expression levels between healthy and infected

plants also in leaves. These findings highlight the need for further studies to pinpoint the molecular mechanisms and the specific effects of OYDV infection on onion physiology, signaling pathways, and/or the sequencing of these molecular targets.

**5. Conclusions**

Our results sustained the hypothesis that OYDV infection induced a water accumulation in onion bulb, indeed, the virus is able to modify the water balance. Consequently, although infected bulb tissues could accumulate water, suggesting at first the production of larger and heavier bulbs, after long-term storage conditions it was evident that this initial increase in weight and size was generally due to less dry matter and more compact bulbs. Indeed, water content has an impact on long-term storage and product quality, increasing the loss rate (92.72 to 95.76% versus an average of 90% in the healthy plant). In addition, these cellular changes affecting bulb quality could be a risk factor for the establishment of secondary pathogen infections, resulting in post-harvest losses.

In conclusion, this study represents the first investigation about the effect of OYDV in terms of water losses and dry matter content on long-term storage, a crucial stage in onion processing cycle, mainly in 'Rossa di Tropea' for which limited information is until now available. The long-term storage is essential for onion crop in covering market demands throughout the year. Thus, this information on the effect of viral infection on marketable product with a determined organoleptic property is crucial for a correct crop management from the field to storage. Therefore, additional studies are needed to further investigate the molecular mechanisms of plant-host (*A. cepa*–OYDV) interaction, valuable for developing strategies to improve onion cropping systems and to enhance the bulb quality.

**Supplementary Materials:** The following supporting information can be downloaded at: https://www.mdpi.com/article/10.3390/horticulturae10010073/s1, Figure S1: Scale of severity of symptoms in OYDV- infected onion plants; Figure S2. Effect of OYDV at 15 and 30 dpi on the highest leaf (cm) (a) and number of leaves in the first-year trial (b). The results are expressed by mean ± the standard deviation. Asterisks indicate the statistical significance resulted in ANOVA analysis ($p < 0.05$) comparing OYDV-infected with healthy samples of a single cultivar in the considered parameters (highest leaf or number of leaves) at 15 and 30 dpi; Figure S3. Effect of OYDV at 15 and 30 dpi on the highest leaf (cm) (a) and number of leaves in the second-year trial (b). The results are expressed by mean ± the standard deviation. Asterisks indicate the statistical significance resulted in ANOVA analysis ($p < 0.05$) comparing OYDV-infected with healthy samples of a single cultivar in the considered parameters (highest leaf or number of leaves) at 15 and 30 dpi; Table S1. Length of fragments obtained from the analysis of onion cultivars/ecotypes by Genemapper v.5.

**Author Contributions:** Conceptualization, A.T. and G.M.; methodology, A.T. and F.S.; formal analysis, G.M., C.L.C. and A.M.; investigation, G.M. and A.T.; data curation, C.L.C., G.M. and A.M.; writing—original draft preparation, C.L.C.; writing-review and editing, C.L.C., A.T., A.M., S.B., F.S. and M.R.A.; supervision, A.T. and F.S. All authors have read and agreed to the published version of the manuscript.

**Funding:** This study was financially supported by (SIR-MIUR grant-SIORTO-RBSI149LD5) funded by the Italian Ministry of University, Education and Research-MIUR, in the frame of Scientific Independence of your Researcher-SIR initiative.

**Data Availability Statement:** Data are contained within the article and supplementary materials.

**Acknowledgments:** The authors wish to thank Adrian Fox, Fera Science Ltd., York (UK) for the scientific revision of the paper.

**Conflicts of Interest:** The authors declare no conflicts of interest.

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
