# Peer review of "Study on Italian Onion Cultivars/Ecotypes towards Onion Yellow Dwarf Virus (OYDV) Infection"

_horticulturae, doi:10.3390/horticulturae10010073_

Round 1
Reviewer 1 Report
Comments and Suggestions for Authors
Congratulations on the manuscript, it's an important and useful piece of work for those working with onions. I will describe some adjustments and questions about the manuscript.
-An important issue for the manuscript should be explained, even to justify the choice of cultivars. What is the probability of the existence of so many ecotypes of the Acquaviva cultivar? Are these plants different in all aspects? Was there any type of selection, or is it simply a natural variability selected due to cultivation in different regions?
- Why wasn't the productivity/average bulb weight evaluated?
Line 100 reads 'Rosa di Tropea (CZ),' while in the rest of the manuscript, it's referred to as 'RC.
- Standardize the cultivar names in Tables 1 and 5 to match the rest of the tables. Ensure consistency in capitalization. Additionally, in Table 1and 4, there's a cultivar named 'Rossa Margherita di Savoia,' whereas in Table 5, it's named 'Rossa Di Margherita,' which is likely the same cultivar.
- Lines 109 to 111 - standardize the writing of numbers. Usually, numbers from zero to ten are written in full, while numbers above ten are expressed as numerals.
- I couldn't understand the inoculation. Make it clear at which stage of development (days) the plants were inoculated.
-Line 259: Presenting the results in PII is confusing. I believe it would be clearer to present the result in the text as PII%.
- Line 268 - Table 4 should be self-explanatory; therefore, I suggest that PII has an asterisk and its meaning, as well as how it was calculated (as explained in line 133), be provided at the bottom of Table 4.
-Lines 274 to 289 and Figure 1: The presentation of the results is quite confusing. It seems that if the results were indeed expressed as mean plus standard deviation, there was no difference in plant growth in both experiments. Thus, the text would be incorrect. Similarly, Figure 1 is confusing and hard to understand where the mean value of each sample is and where the standard deviation extends to. Moreover, combining results from different cultivars is not very informative since there are susceptible and resistant cultivars in the evaluated group. Therefore, this discussion should be conducted by cultivar, showcasing the virus's effect on the growth of each of the evaluated cultivars. Just as was done for the variables bulb long-term storage and water losses.
- It wasn't discussed why Acquaviva was the most susceptible in the first year and in the second year had almost half the infection rate compared to Pera Sanguigna di Peschici. Additionally, the discussion did not cover why Acquaviva ecotype 2 showed nearly the same infection rate as the resistance standard, while ecotype 7 was the susceptibility standard. Furthermore, in the second experiment, a new ecotype that wasn't evaluated in the first year (Rossa di Tropea VV) was introduced, while Acquaviva ecotype 2 wasn't reassessed.
It would be very interesting to include the different ecotypes of the 'Acquaviva' cultivar in the diversity study, given the demonstrated significant variability in virus resistance.
Author Response
Reviewer 1
We would like to thank the reviewer for all the comments. We implemented the manuscript where requested.
-An important issue for the manuscript should be explained, even to justify the choice of cultivars. What is the probability of the existence of so many ecotypes of the Acquaviva cultivar? Are these plants different in all aspects? Was there any type of selection, or is it simply a natural variability selected due to cultivation in different regions?
Thank you for the comment above; indeed, the great variability observed in the Acquaviva ecotypes surprised us as well. There was not a selection but a natural variability. Onion is known to be allogamous and probably that specific cultivar is not in a stable Hardy Weinberg equilibrium.
- Why wasn't the productivity/average bulb weight evaluated?
The aim of the study was to have a preliminary evaluation of different Italian ecotypes and cultivars to retrieve potential tolerance traits. The experimental trials were conducted in protected conditions and in pots. For sure, this was the first step, and in the future, we are considering the evaluation of data concerning productivity establishing experimental in field parcels following the conventional production procedures.
- I couldn't understand the inoculation. Make it clear at which stage of development (days) the plants were inoculated.
We included further information in the text. The inoculation was performed soaking the needle of a syringe in a 50 ml tube (where was collected a plant crude sap obtained by maceration of infected plant material in phosphate buffer (0.1 M; 1:5 w:v)); puncturing ten times along two leaves each plant. The needle was soaked before every single puncture. Plantlets were obtained starting from seeds, transplanted after 4 four months after sowing and the inoculation was performed one month after the transplant.
- It wasn't discussed why Acquaviva was the most susceptible in the first year and in the second year had almost half the infection rate compared to Pera Sanguigna di Peschici. Additionally, the discussion did not cover why Acquaviva ecotype 2 showed nearly the same infection rate as the resistance standard, while ecotype 7 was the susceptibility standard. Furthermore, in the second experiment, a new ecotype that wasn't evaluated in the first year (Rossa di Tropea VV) was introduced, while Acquaviva ecotype 2 wasn't reassessed.
It would be very interesting to include the different ecotypes of the 'Acquaviva' cultivar in the diversity study, given the demonstrated significant variability in virus resistance.
We completely agree that acquaviva could represent a very interesting susceptible cultivar to be studied in deep, in view of its high variability, that partially could justify a different PII% in the second-year trail. Our main aim was focus in retrieving tolerant traits and we will include this cultivar in future studies concerning plant-pathogen interactions, despite its intrinsic high variability could represent a limit. The data reported for Acquaviva 2 ecotype was wrong (it was a typo the correct PII% was 77.9 and not 7.9), thus it was excluded as well as other susceptible cultivars sharing same PII% values.
‘Rossa di Tropea’ (VV) was included as to double check and confirm the tolerance traits observed in this cultivar, reported in literature to be susceptible to OYDV. In addition, it was demonstrated that this cultivar is in hardy Weinber equilibrium (Puccio et al., 2022, 10.3390/cells11071100) confirmed also in the analysis included in this study.

Reviewer 2 Report
Comments and Suggestions for Authors
Dear Authors, the subject of the manuscript is quite interesting. However, I do not understand that with 7 coauthors and an English reader you have so many basic errors in the text and in interpretation. This manuscript is like to being the first written version to circulate among coauthors for correction. There is lot to say fort a large part of you results and all this is mentioned in the attached file. Due to the short time for reviewing it could be that more change needs to be made out of the one I noted.

No specific comment but numbers are miswritten (x,y in place of x.y) and some words are missing to better understand the meaning of the text
Author Response
We would like to thank the reviewer for all the comments. We really appreciate the effort done to allow the improvement of the manuscript. We accepted all the suggestions and amended the text accordingly. Please the attached file.

Reviewer 3 Report
Comments and Suggestions for Authors
In this MS the authors describe their results concerning the evaluation of resistant and susceptible onion cultivars againsts OYDV based on the distinct symptom severity observed among cultivars/ecotypes, the effect of OYDV infection on water loss and bulb long-term storage. Furthermore the phylogenetic relationship of these cultivars/ecotypes was investigated through an SSR analysis. Finally, the interaction of OYDV with each cultivar studied was evaluated through the expression of isoforms of EIF4e and EIF4g. Since this is one of the first studies that investigate the interraction between onion cultivars and OYDV, its results can be used to further enlighten the onion/OYDV pathosystem and to identify new types of genetic resistance. Overall the MS is well written and the experiments well designed and performed, therefore I recommend publication of the MS after minor revision.
My only addition would be to include a figure (in section 3.1.1) of the different symptoms (0 to 5) that were used to assess symptom severity. Some minor comments can be found in the attached file.

Author Response
Reviewer 3
In this MS the authors describe their results concerning the evaluation of resistant and susceptible onion cultivars againsts OYDV based on the distinct symptom severity observed among cultivars/ecotypes, the effect of OYDV infection on water loss and bulb long-term storage. Furthermore the phylogenetic relationship of these cultivars/ecotypes was investigated through an SSR analysis. Finally, the interaction of OYDV with each cultivar studied was evaluated through the expression of isoforms of EIF4e and EIF4g. Since this is one of the first studies that investigate the interraction between onion cultivars and OYDV, its results can be used to further enlighten the onion/OYDV pathosystem and to identify new types of genetic resistance. Overall the MS is well written and the experiments well designed and performed, therefore I recommend publication of the MS after minor revision.
My only addition would be to include a figure (in section 3.1.1) of the different symptoms (0 to 5) that were used to assess symptom severity. Some minor comments can be found in the attached file.
We would like to thank the reviewer for the work done to improve the manuscript. We accepted all the comments and suggestion reported in the attached file (including the shifting of Table 5 as supplementary Figure 1). In addition, was included a Supplementary figure 1 reporting pictures of the symptom’s severity used for the PII% assessment.
Round 2
Reviewer 2 Report
Comments and Suggestions for Authors
dear authors, there is still few editing corrections to be made, see attached file.

no comment
Author Response
Thank you again for the revision, we accepted all the correction and revised the manuscript accordingly.
Kind regards